# The Cost-Effectiveness of Increased Yogurt Intake in Type 2 Diabetes in Japan

**DOI:** 10.3390/nu17142278

**Published:** 2025-07-09

**Authors:** Ryota Wakayama, Michihiro Araki, Mieko Nakamura, Nayu Ikeda

**Affiliations:** 1National Institute of Health and Nutrition, National Institutes of Biochemical Innovation, Health and Nutrition, 3-17 Senriokashimmachi, Settsu 566-0002, Osaka, Japan; 2Meiji Co., Ltd., 2-2-1 Kyobashi, Chuo-ku 104-8306, Tokyo, Japan; 3Department of Pharmaceutical Sciences, College of Pharmaceutical Sciences, Ritsumeikan University, 1-1-1 Noji-higashi, Kusatsu 525-8577, Shiga, Japan

**Keywords:** Markov model, simulation, data science, Japanese diet, dairy

## Abstract

**Background/Objectives**: A healthy diet helps prevent noncommunicable diseases, and dairy is an essential part of this diet. Multiple meta-analyses have shown an inverse association between yogurt intake and type 2 diabetes (T2D). This study aimed to develop a simulation model and evaluate the medical and economic effects of increased yogurt intake on T2D. **Methods**: It predicted the T2D incidence rate, T2D mortality rate, and national healthcare expenditures (NHE) over 10 years using a Markov model for the Japanese population aged 40–79 years. **Results**: By increasing yogurt intake to 160 g/day or 80 g/day, the incidence rate of T2D decreased by 16.1% or 5.9%, the T2D-related mortality rate decreased by 1.6% or 0.6%, and the NHE was predicted to decrease by 2.4% and 0.9%, respectively. **Conclusions**: Increasing yogurt intake may be an effective strategy to prevent T2D and reduce NHE.

## 1. Introduction

Diabetes is a major health and economic issue in Japan, representing one of the most critical lifestyle-related diseases [1,2,3,4,5]. According to the National Health and Nutrition Survey in 2016, the number of people with diabetes (defined as having a hemoglobin A1c [HbA1c] level of ≥6.5% or having been diagnosed with diabetes) and those at risk (with HbA1c levels between 6.0% and 6.5%) is approximately 20 million in Japan [1,6]. The number of people with diabetes has shown a continuous upward trend [6]. Diabetes is associated with complications, such as diabetic neuropathy, retinopathy, and nephropathy [7,8,9], and it is a risk factor for cardiovascular diseases, such as myocardial infarction and stroke [10]. Additionally, it increases the risk of developing dementia [11] and cancers [12]. Furthermore, diabetes is the leading cause of new dialysis initiation [13]. In Japan, national healthcare expenditures (NHEs) due to diabetes, including both inpatient and outpatient care, amount to 9.4 trillion US dollars (USD) annually [14]. Due to its profound impact on quality of life, socioeconomic vitality, and social resources, the implementation of appropriate countermeasures is important [15,16]. The implementation of ongoing and comprehensive interventions to address diabetes and its associated complications is essential for promoting public health outcomes [17,18,19]. Thus, reducing the incidence of diabetes has been designated as a key objective by the Ministry of Health, Labour and Welfare [1].

Type 2 diabetes (T2D) is closely associated with diet. Many cases of T2D develop because of a combination of multiple genetic factors related to decreased insulin secretion and resistance, along with lifestyle factors, such as overeating and lack of exercise [2]. Therefore, treating T2D with a healthy diet is necessary. A healthy diet helps protect against noncommunicable diseases, such as diabetes and cardiovascular diseases [20]. The intake of dairy products is recommended by many dietary guidelines worldwide [21]. In Japan, several guidelines for a healthy diet exist, including the Japanese Food Guide Spinning Top [22,23], Health Japan 21 [24], and Recommendations for Extending Healthy Life Expectancy Based on Cross-Disease Evidence [25], all of which recommend the intake of dairy products. Therefore, dairy products are essential components of a healthy diet. A previous study indicated that dietary patterns, including dairy products, were associated with a lower risk of impaired fasting glucose, a precursor of T2D [26].

Dairy product intake has positive effects on various lifestyle-related diseases, including T2D [27,28,29,30,31,32,33], overweight and obesity [34,35], and hypertension [36,37,38]. Notably, the inverse association between the intake of yogurt, which is a type of fermented milk, and T2D has been consistently observed in multiple meta-analyses [27,28,29,30,31,32,39]. A dose–response meta-analysis showed that consuming yogurt resulted in a lower relative risk for T2D (0.93 per 50 g/day) than that by other types of dairy products (relative risk for total dairy, 0.97 per 200 g/day; for low-fat dairy, 0.96 per 200 g/day) [39]. This indicates that among dairy products, yogurt has a significant impact on reducing the risk of T2D. In 2024, the US Food and Drug Administration (FDA) announced that regular intake of yogurt, at least two cups per week, may reduce the risk of T2D [40]. In Japan, the Japanese Food Guide Spinning Top recommends two servings (SV) of dairy products as daily intake, with one SV defined as the amount equivalent to approximately 100 mg of calcium derived from the main ingredient [22,23]. According to the nutritional information for yogurt listed in the food composition tables in Japan [41], if the recommended amount of dairy products is met solely with yogurt, two SV is equivalent to 160 g yogurt. According to the National Health and Nutrition Survey in 2019, the mean intake of total dairy products per day for individuals aged ≥20 years is 110.7 g [6]. A previous study showed that the mean daily yogurt intake was 33.2 g [42]. Despite the benefits of yogurt intake for T2D, its intake is still lower compared to the recommended level.

Promoting yogurt intake can be an effective strategy for curbing the increase in NHE and extending healthy life expectancy. A previous simulation study conducted in Iran demonstrated that the intake of dairy products, including yogurt, could reduce NHE-related T2D [43]. Therefore, the consumption of yogurt, a type of dairy, may reduce the risk of T2D and associated NHE in Japan. However, the economic impact of increased yogurt intake on T2D remains unclear in Japan. Although a previous study simulated the medical economic effects of dairy intake, including yogurt [43], none have focused on the healthcare economic effects of yogurt alone. Therefore, this study sought to develop a simulation model to assess the medical and economic effects of increased yogurt intake in Japanese patients with T2D.

## 2. Materials and Methods

### 2.1. Modeling Framework

To evaluate the medical and economic impacts of increased yogurt intake, we focused on T2D incidence and NHE in Japan. To model the transitions between multiple health states over time, a Markov cohort simulation model was selected using TreeAge Pro Healthcare 2024 (TreeAge Software, Williamstown, MA, USA) [44].

A closed cohort was simulated for the Japanese population aged 40–79 years in 2019 over a period of 10 years. The reason for targeting adults aged ≥40 years in this study is that T2D incidence is low among adults aged <40 years in Japan [6]. The year 2019 was selected as the baseline year because it was the latest year for which all the necessary data for the Markov simulation was available. To minimize the impact of long-term social changes and ensure reliable results, we selected a simulation period of 10 years.

In this study, the model included four mutually exclusive states: “Healthy,” “Chronic T2D,” “Death from T2D,” and “Death from other causes” (Figure 1). The “Healthy” state represented individuals who had never been diagnosed with T2D; the “Chronic T2D” state represented individuals who had been diagnosed with T2D; the “Death from T2D” state represented individuals who died from T2D; and “Death from other causes” state represented individuals who died from causes other than T2D. Complications related to T2D were not included in this model owing to insufficient epidemiological and healthcare expenditure data in Japan.

At the beginning of the Markov simulation, individuals were divided into “Healthy” and “Chronic T2D” states based on T2D prevalence. Annually, individuals in the cohort transitioned between the four states based on the transition probabilities. Individuals in the “Healthy” state had three possible outcomes: they could develop T2D and transition to the “Chronic T2D” state, die from causes other than T2D and transition to the “Death from other causes” state, or remain in the “Healthy” state. Once in the “Chronic T2D” state, it is assumed that individuals would not revert to the “Healthy” state. Thus, individuals in the “Chronic T2D” state could either remain in that state, die from T2D, and transition to the “Death from T2D” state or die from causes other than T2D and transition to the “Death from other causes” state.

### 2.2. Scenarios

The Japanese Food Guide Spinning Top recommends the consumption of dairy products equivalent to 200 mg of calcium per day. According to the food composition tables in Japan [41], yogurt equivalent to 200 mg of calcium corresponds to 160 g. In this study, we assumed that the recommended dairy product intake would be achieved entirely through yogurt consumption. We examined two scenarios in which yogurt intake was increased to 160 g/day (scenario 1) and 80 g/day (scenario 2). Scenario 1 corresponds to two SV in the Japanese Food Guide Spinning Top, whereas scenario 2 corresponds to one SV. Given that the serving size of yogurt in Japan is 80–100 g [45,46], we assumed an immediate increase in yogurt intake of 80 or 160 g/day. To establish a baseline scenario for comparison, we set up a scenario in which the mean yogurt intake in 2019 continued for 10 years. The health and economic impacts of increased yogurt intake were evaluated by comparing the predicted T2D incidence, T2D mortality, and NHE between the baseline scenario and the two scenarios with increased yogurt intake.

### 2.3. Input Parameters

The input parameters used in the Markov simulation were obtained from public databases and the academic literature (Table 1). Because data on yogurt intake alone are not available from the National Health and Nutrition Survey in Japan, the intake of fermented milk and probiotic beverages, including yogurt, was substituted for yogurt intake. The mean yogurt intake substituted by the intake of fermented milk and probiotic beverages in 2019, broken down by sex and age, is shown in Table 2. Probabilities among the states were determined using the incidence rates of T2D, all-cause mortality rates, relative risk of T2D associated with yogurt intake, and mortality rates of T2D (Table 2). The transition probability from “Healthy” to “Chronic T2D” was calculated by multiplying the incidence rate of T2D by the relative risk for T2D associated with increased yogurt intake. The relative risk for T2D per 50 g, as shown by a dose–response meta-analysis [39] was extrapolated to estimate the reduction in risk associated with increased yogurt intake. Additionally, the transition probabilities from “Healthy” or “Chronic T2D” to “Death from other causes” were calculated by subtracting the mortality rate of T2D from the all-cause mortality rate. Medical costs were allocated to outpatient care cost, including dispensing fees, for the transition from “Healthy” state to “Chronic T2D” state (Table 3). In this study, we did not incorporate inpatient care costs for T2D because of the lack of data necessary for the simulation. The NHE was converted from Japanese yen (JPY) to USD based on the annual average exchange rate in 2019 (1 USD = 109.01 JPY) published by the International Monetary Fund [47]. These NHEs were discounted at an annual rate of 2%, following a guideline for economic evaluation [48].

### 2.4. Sensitivity Analysis

Multiple deterministic one-way sensitivity analyses were performed to evaluate the impact of parameter uncertainty on the model results. The parameters examined in this analysis included the incidence rate of T2D, prevalence rate of T2D, T2D and all-cause mortality rates, relative risk of T2D, and discount rate. For the incidence rate of T2D, the prevalence rate of T2D, T2D, and all-cause mortality rates and relative risk for T2D, 95% confidence intervals were used (Table 2). For the discount rate, a range of 0–4% was used.

## 3. Results

### 3.1. Projected Incidence, Mortality, and NHE on Baseline Scenario

In 2019, the number of Japanese people aged 40–79 years was 66,958,000. Both men and women in the 40–49-year age group had the highest population. The projected T2D incidence, deaths due to T2D, and NHE for T2D at the end of the 10-year simulation in the baseline scenario are shown in Table 4. If the mean daily yogurt intake in 2019 was maintained for over 10 years, a total of 2,223,068 new T2D cases (3.3%) would occur. The T2D incidence ranged from 192,728 to 389,553 in men and from 192,413 to 273,451 in women. Additionally, the cumulative number of deaths due to T2D was 4907 (0.007%). The cumulative NHE was USD 55,546,282,937. The NHE ranged from USD 3,917,130,410 to USD 12,580,641,487 in men and from USD 1,827,846,469 to USD 9,489,753,708 in women.

### 3.2. Health Gains by Increased Yogurt Intake

Table 5 shows the prevention of T2D incidence and death due to increased yogurt intake compared with the baseline scenario. The incidences of T2D prevention were 356,866 (16.1%) in scenario 1 and 131,108 (5.9%) in scenario 2. In scenario 1, the incidence of T2D prevention ranged from 30,180 to 65,104 cases in men and from 31,869 to 40,672 cases in women. In scenario 2, the incidence of T2D prevention ranged from 10,471 to 26,075 cases in men and from 9056 to 12,472 cases in women. In scenarios 1 and 2, the highest incidence of T2D was observed in men aged 50–59 years and women aged 60–69 years. Additionally, death from T2D was prevented in 77 (1.6%) patients in scenario 1 and 27 (0.6%) patients in scenario 2.

### 3.3. Saved NHE by Increased Yogurt Intake

Table 6 shows the projected cumulative savings in NHE due to the prevention of T2D by increased yogurt intake. The projected cumulative savings for the entire cohort were USD 1324 million (2.4% of the baseline scenario) in scenario 1 and USD 481 million (0.9% of the baseline scenario) in scenario 2. In scenario 1, savings ranged from USD 153,401,134 to USD 264,395,111 in men and from USD 77,293,678 to USD 160,539,759 in women. In scenario 2, savings ranged from USD 53,252,481 to USD 101,428,663 in men and from USD 29,807,938 to USD 49,257,693 in women. In both scenarios, the highest savings were observed in both men and women aged 60–69 years.

### 3.4. Sensitivity Analyses

Table 7A,B show the results of the one-way sensitivity analyses. In both scenarios 1 and 2, the greatest uncertainty in the projections using the Markov model was associated with the relative risk for T2D. Owing to the uncertainty in the relative risk of T2D, the range of predicted cumulative savings for T2D were USD 854,572,352 in scenario 1 and USD 356,030,915 in scenario 2. In Table 7, “Low” and “High” indicate the lower and upper bounds, respectively, of the uncertainty in predictions based on the input parameters used in the simulations, and “Expected Values” indicate the predicted cumulative savings of NHE for T2D.

## 4. Discussion

This is the first study to evaluate the medical and economic effects of yogurt intake on T2D in a Japanese population. T2D incidence and NHE were targeted to assess the medical and economic impacts of yogurt intake on T2D. The mechanisms by which yogurt intake affects the onset of T2D have not been fully understood. However, previous studies may help explain some of these mechanisms. Components such as minerals, vitamins, whey proteins, and certain fatty acids found in dairy products may help to reduce the risk of T2D [51,52]. Calcium and vitamin D have supportive roles in pancreatic beta-cell function and glucose metabolism [52]. Additionally, whey protein may exert insulinotropic and glucose-lowering effects [53]. Vitamin K2 in yogurt is inversely associated with T2D risk [54]. Additionally, yogurt may influence the gut microbiota through probiotics [55]. Lactic acid bacteria in yogurt reduces chronic inflammation and subsequent insulin resistance [56]. Chronic inflammation is widely recognized as a principal factor contributing to the impairment of glucose metabolism, primarily through its role in promoting insulin resistance, which in turn facilitates the onset of metabolic disorders [57,58,59]. The reduction in chronic inflammation through lactic acid bacteria consumption may help mitigate insulin resistance, thereby potentially improving glucose metabolism [56]. Although the exact mechanisms remain unclear, multiple meta-analyses have shown the inverse correlation between increased yogurt intake and the risk of developing T2D [27,28,29,30,31,32,39,60].

Following the recommendations of the Japanese Food Guide Spinning Top has been suggested to improve diet quality [61] and potentially prevent obesity and all-cause mortality [62,63,64]. Our simulation results suggest that increasing yogurt intake to the recommended amounts of dairy products can reduce T2D incidence by 16.1% over 10 years. Although the population aged 40–79 years in Japan is larger for women than for men, the effect of yogurt intake on T2D incidence was greater for men than for women. This may be because men have a higher T2D prevalence and T2D incidence than women, making them more susceptible to developing T2D, which could influence the positive effects of yogurt intake. Furthermore, reducing T2D incidence could lead to a 1.6% decrease in deaths by T2D. These results suggest that yogurt intake may help maintain a healthy state in the Japanese population. From an economic perspective, this could potentially reduce NHEs due to T2D by 2.4%. Therefore, increasing yogurt intake can reduce T2D incidence and consequently reduce NHEs. Taking the recommended amounts of dairy products, in the Japanese Food Guide Spinning Top, can prevent T2D and reduce the medical economic burden caused by T2D. Our results are consistent with those of previous studies in Iran, which showed that increasing dairy intake led to T2D prevention and medical economic benefits [43]. A previous study in Iran found that dairy consumption generated medical economic effects of USD 2400 million over 10 years and USD 11,191 million over 20 years. Considering the effects of dairy on T2D, increasing the intake of dairy products, including yogurt, may produce greater medical and economic benefits than those suggested by our simulation results. Additionally, extending the simulation period beyond 10 years may yield greater medical and economic benefits.

In Japan, the goals for diabetes countermeasures are divided into three stages: “prevention of diabetes onset,” “prevention of complications through appropriate diabetes treatment,” and “prevention of organ damage and improvement of life prognosis due to complications” [1]. This study focused solely on the primary prevention of T2D and conducted simulations; we only input outpatient costs into the Markov simulation as medical costs. However, T2D has numerous complications, such as neuropathy, retinopathy, and diabetic chronic kidney disease (CKD) [7,8,9]. For example, patients with CKD may require treatment through renal dialysis. In Japan, the primary underlying disease for renal dialysis is the progression of diabetic nephropathy. The medical costs of renal dialysis amount to approximately USD 15 billion annually [65], which poses a significant economic burden. Additionally, T2D treatment may also require hospitalization, and inpatient care costs due to T2D amount to approximately USD 1223 million annually in the Japanese population aged 40–79 years [14]. However, we could not incorporate inpatient care costs due to T2D or medical costs related to diabetes complications into the simulation owing to a lack of necessary data for the simulation in this study. Furthermore, yogurt intake has been suggested to have beneficial effects on diseases other than T2D, such as obesity [34,35] and hypertension [36,37,38]. The consumption of dairy products can prevent cardiovascular diseases and generate economic benefits [43,66]. Japanese studies have also shown that higher yogurt intake is associated with better control of cardiovascular risk factors, including CKD [67]. Thus, if the effects of yogurt intake on diseases that reduce the risk of T2D complications were included in the simulation, the medical and economic benefits of yogurt intake would be even more significant.

This study conducted simulations by considering the intake of fermented milk and probiotic beverages, including yogurt, from the National Health and Nutrition Survey in Japan as yogurt intake, as data for yogurt intake alone were not available. According to the National Health and Nutrition Survey in 2019, the mean intake of fermented milk and probiotic beverages, including yogurt, in Japanese populations aged 40–79 years ranges from 24.5 to 51.4 g/day (Table 2) [6]. Additionally, based on the annual shipment volume of yogurt alone in Japan in 2019 (637.7 million kg/year) [68], the per capita intake of yogurt alone was calculated to be 13.8 g/day. In contrast, the median intake of fermented milk and probiotic beverages, including yogurt, in Japan is 0 g/day [6]. The relationship between the mean and median intake of fermented milk and probiotic beverages, including yogurt, indicates that the daily intake of many people is 0 g, which shows a highly skewed distribution. These findings suggest that yogurt intake may vary significantly among individuals in Japan and that the actual intake of yogurt alone may be even lower than the mean intake of fermented milk and probiotic beverages. Therefore, the medical and economic effects of actual yogurt intake on T2D may be greater than the simulation results.

From the perspective of T2D prevention and healthcare cost reduction, it may be necessary to implement initiatives to promote yogurt intake. Specifically, policymakers can encourage food manufacturers to reformulate their products to be healthier, or food manufacturers can voluntarily reformulate their products to be healthier. One mechanism that promotes product reformulation is the nutritional profiling system (NPS) [69]. The NPS evaluates the nutritional value of food items by scoring or classifying them based on the amount of nutrients and other components [70,71]. The Global Access to Nutrition Index has motivated food manufacturers to reformulate their products using the NPS [72]. In Japan, several NPSs have been developed considering public health problems [73,74,75]. Additionally, NPSs that promote dairy consumption have been developed [76,77]. Utilizing these systems to increase yogurt consumption can help address public health problems.

### Limitations

This study has some limitations. First, to the best of our knowledge, there are no studies on the relative risk for T2D associated with increased yogurt intake in the Japanese population. Therefore, this study used the values of the relative risk of T2D associated with increased yogurt intake obtained from a meta-analysis in our simulation. The results of the sensitivity analyses indicated that the relative risk for T2D was the greatest factor of uncertainty. Depending on the values of the relative risk of T2D associated with increased yogurt intake in the Japanese population, the resulting healthcare economic effects may vary according to our simulation. Second, this study targeted the Japanese population aged 40–79 years and set the simulation period at 10 years. Population changes over the 10-year period were not considered in the model. Therefore, our model has certain limitations, and the results obtained are specific to our model. Third, we did not include the cost of purchasing yogurt in our model. The prices of dairy products in Japan continue to increase [78]. Maintaining yogurt intake over a 10-year period may require the cooperation and support of policymakers and food manufacturers. Thus, food affordability may be important for continued yogurt consumption. Fourth, we used the intake of fermented milk and probiotic beverages, which include yogurt, for the simulation because the intake of yogurt alone in Japan has not been published. Therefore, the results obtained from the simulation may vary depending on actual intake of yogurt alone. Fifth, only outpatient care costs were considered as medical costs in the simulation. However, medical costs related to T2D also include diabetes complications and inpatient care costs. These costs were not considered in the simulation model because of the lack of necessary data. Finally, this study did not distinguish between sweetened and unsweetened yogurt owing to the lack of data on the intake of each type. In addition, due to the lack of necessary data, it was not possible to incorporate variables such as yogurt type and additives like non-sugar sweeteners into the simulation. The FDA health claims support the health benefits of yogurt as a food item, regardless of its fat or sugar content [40]. However, consuming sweetened yogurt may increase sugar intake, potentially negatively affecting T2D risk.

## 5. Conclusions

A Markov model simulation was conducted to evaluate the impact of yogurt intake on T2D and NHEs in Japan. The results of this study indicate that increasing yogurt intake may serve an effective preventive strategy against T2D in Japan, with the potential to reduce both T2D incidence and NHEs, underscoring the potential role of yogurt in public health. However, current yogurt intake in Japan is insufficient compared with the recommended amount, highlighting the critical need for public health initiatives to promote yogurt intake. To enhance public health outcomes, policymakers and food manufacturers should focus on improving access to healthy foods, food affordability, and nutritional education. This research supports future public health strategies highlighting the economic benefits of adopting dietary recommendations.

## Figures and Tables

**Figure 1 nutrients-17-02278-f001:**
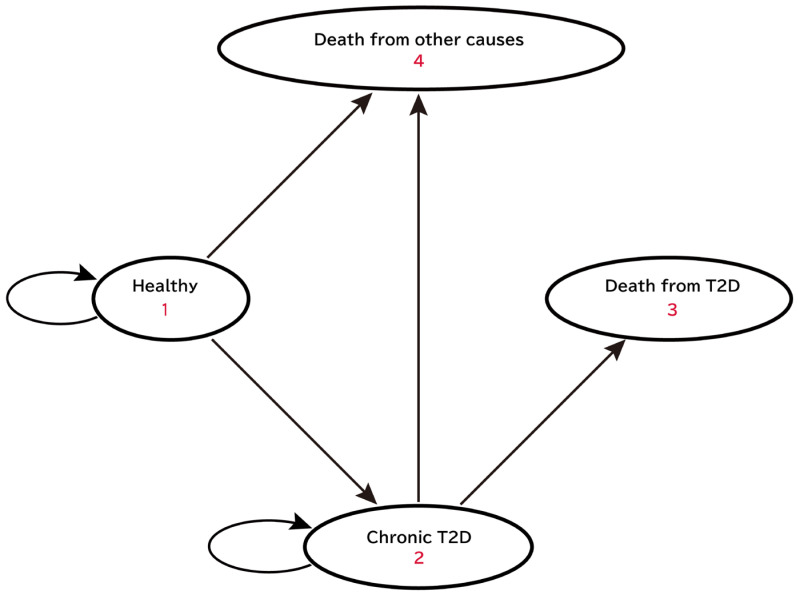
State transition diagram of the Markov model. The model includes four states: (1) Healthy, (2) Chronic T2D, (3) Death from T2D, and (4) Death from other causes. The ellipses represent states in the simulation. The straight arrows indicate the direction of transitions between states, and the curved arrows indicate remaining in the same state. T2D, type 2 diabetes.

**Table 1 nutrients-17-02278-t001:** Data sources of the input parameters for the Markov simulation.

Input Parameters	Data Source
Total population	Population Estimate, 2019 [49]
Mean yogurt intake (g/day)	National Health and Nutrition Survey in Japan, 2019 [6]
Prevalence rates of T2D	Global Burden of Disease Study 2019 [50]
Incidence rates of T2D	Global Burden of Disease Study 2019 [50]
Mortality rates of T2D	Global Burden of Disease Study 2019 [50]
All-cause mortality rates	Global Burden of Disease Study 2019 [50]
Relative risk for T2D associated with yogurt intake	Dose–response meta-analysis of cohort studies [39]
National healthcare expenditures	Survey on Medical Insurance Benefits, 2019 [14]

The mean yogurt intake values were substituted with the values for fermented milk and probiotic beverages, including yogurt, from the National Health and Nutrition Survey in Japan in 2019. T2D, type 2 diabetes.

**Table 2 nutrients-17-02278-t002:** Input parameters for the Markov simulation.

Sex, Age (Years)	Total Population [49]	Yogurt Intake (Mean) [6]	T2D Incidence [50]	T2D Prevalence [50]	T2D Mortality [50]	All-Cause Mortality [50]	Relative Risk for T2D [39]
	No.	g/day	per 100,000	per 100,000	per 100,000	per 100,000	per 50 g
Men							
40–49	9,374,000	24.5	386.9 (289.6–498.6)	5716.5 (4728.7–6726.4)	1.1 (1.0–1.3)	150.6 (147.9–153.4)	0.93 (0.89–0.97)
50–59	8,161,000	31.1	520.9 (406.6–658.2)	10,032.7 (8734.0–11,449.3)	3.5 (3.2–3.8)	390.6 (383.1–398.5)	
60–69	7,930,000	34.4	517.9 (385.1–661.3)	15,065.7 (13,385.8–16,975.4)	7.6 (7.0– 8.2)	996.9 (979.4–1015.4)	
70–79	7,333,000	40.9	317.7 (224.6–418.3)	18,541.9 (16,590.8–20,761.4)	17.0 (15.5–18.4)	2579.2 (2538.1–2622.3)	
Women							
40–49	9,147,000	34.1	224.1 (163.3–295.4)	3310.0 (2697.6–3992.1)	0.3 (0.3–0.4)	88.2 (86.6–90.0)	0.93 (0.89–0.97)
50–59	8,117,000	44.7	327.5 (244.7–430.3)	5957.5 (5070.1–7038.2)	0.9 (0.8–1.0)	201.1 (197.4–205.1)	
60–69	8,302,000	47.3	365.9 (262.1–483.7)	9303.4 (8126.7–10,724.5)	2.7 (2.5–3.0)	439.3 (432.3–446.8)	
70–79	8,594,000	51.4	303.0 (220.4–395.8)	12,210.5 (10,786.8–13,856.9)	8.9 (7.5–9.9)	1205.0 (1187.4–1223.8)	

The mean yogurt intake values were substituted with the values for fermented milk and probiotic beverages, including yogurt, from the National Health and Nutrition Survey in Japan in 2019. T2D, type 2 diabetes. Values in parentheses indicate 95% confidence intervals.

**Table 3 nutrients-17-02278-t003:** National healthcare expenditures for diabetes in Japan (2019).

Sex, Age (Year)	Outpatient (USD) [14]
Men	
40–49	320,339,533
50–59	599,793,593
60–69	1,085,266,222
70–79	1,452,129,242
Women	
40–49	149,412,727
50–59	284,612,996
60–69	620,583,959
70–79	985,138,127

USD, US dollars.

**Table 4 nutrients-17-02278-t004:** Projected cumulative T2D incidence, death by T2D, and national healthcare expenditures for T2D over 10 years from 2019 under the baseline scenario of mean yogurt intake remaining at the levels of 2019.

Sex, Age (Years)	Population in 2019	T2D Incidence		T2D Death		National Healthcare Expenditures for T2D
	No.	No.	%	No.	%	USD
Men						
40–79	32,798,000	1,288,049	3.9	3618	0.011	34,318,611,964
40–49	9,374,000	341,551	3.6	76	0.001	3,917,130,410
50–59	8,161,000	389,553	4.8	345	0.004	6,781,869,201
60–69	7,930,000	364,215	4.6	998	0.013	11,038,970,866
70–79	7,333,000	192,729	2.6	2199	0.030	12,580,641,487
Women						
40–79	34,160,000	935,019	2.7	1288	0.004	21,227,670,973
40–49	9,147,000	192,413	2.1	13	0.000	1,827,846,469
50–59	8,117,000	243,340	3.0	54	0.001	3,281,760,296
60–69	8,302,000	273,451	3.3	242	0.003	6,628,310,500
70–79	8,594,000	225,815	2.6	980	0.011	9,489,753,708

T2D, type 2 diabetes; No., number; USD, US dollars.

**Table 5 nutrients-17-02278-t005:** Projected cumulative T2D incidence and death prevented by increased yogurt intake compared with the baseline scenario over 10 years from 2019.

Sex, Age (Years)	T2D Incidence				T2D Death			
	Scenario 1		Scenario 2		Scenario 1		Scenario 2	
	No.	%	No.	%	No.	%	No.	%
Men								
40–79	214,762	16.7	85,310	6.6	57	1.6	21	0.6
40–49	60,089	17.6	26,001	7.6	3	4.0	1	1.7
50–59	65,104	16.7	26,075	6.7	10	3.0	4	1.2
60–69	59,389	16.3	22,762	6.2	21	2.1	8	0.8
70–79	30,180	15.7	10,471	5.4	23	1.0	8	0.4
Women								
40–79	142,104	15.2	45,799	4.9	20	1.6	6	0.5
40–49	31,869	16.6	12,286	6.4	0	3.7	0	1.4
50–59	37,033	15.2	11,985	4.9	2	2.9	1	0.9
60–69	40,672	14.9	12,472	4.6	5	2.1	2	0.6
70–79	32,530	14.4	9056	4.0	13	1.3	4	0.4

T2D, type 2 diabetes; No., number; USD, US dollars.

**Table 6 nutrients-17-02278-t006:** Projected cumulative savings of national healthcare expenditures for type 2 diabetes by increased yogurt intake compared with the baseline scenario over 10 years from 2019.

Sex, Age (Years)	Scenario 1		Scenario 2	
	USD	%	USD	%
Men				
40–79	831,229,401	2.4	326,125,482	1.0
40–49	177,821,247	4.5	76,994,075	2.0
50–59	235,611,909	3.5	94,450,264	1.4
60–69	264,395,111	2.4	101,428,663	0.9
70–79	153,401,134	1.2	53,252,481	0.4
Women				
40–79	493,253,453	2.3	155,066,397	0.7
40–49	77,293,678	4.2	29,807,938	1.6
50–59	107,282,621	3.3	34,737,231	1.1
60–69	160,539,759	2.4	49,257,693	0.7
70–79	148,137,396	1.6	41,263,535	0.4

USD, US dollars.

**Table 7 nutrients-17-02278-t007:** (**A**) One-way sensitivity analyses on projected cumulative savings for T2D under scenario 1. (**B**) One-way sensitivity analyses on projected cumulative savings for T2D under scenario 2.

(A)			
Sex, Parameters	Low	High	Expected Values
	USD	USD	USD
Men			831,229,401
Relative risk for T2D	377,096,111	1,231,668,464	
Incidence rate of T2D	626,250,543	1,058,550,712	
Prevalence rate of T2D	728,377,946	954,373,842	
Discount rate	743,686,382	933,396,936	
All-cause mortality rate	830,416,028	832,003,471	
T2D-related mortality rate	831,215,444	831,243,340	
Women			493,253,453
Relative risk for T2D	225,217,896	725,874,965	
Incidence rate of T2D	360,965,491	644,994,899	
Prevalence rate of T2D	424,586,181	572,494,735	
Discount rate	441,015,042	554,242,622	
All-cause mortality rate	492,986,196	493,503,482	
T2D-related mortality rate	493,245,445	493,259,065	
(**B**)			
**Sex,** **Parameters**	**Low**	**High**	**Expected Values**
	**USD**	**USD**	**USD**
Men			326,125,482
Relative risk for T2D	143,285,100	499,316,015	
Incidence rate of T2D	246,154,824	414,928,697	
Prevalence rate of T2D	285,414,302	375,123,507	
Discount rate	291,767,506	366,224,063	
All-cause mortality rate	325,824,236	326,412,104	
T2D-related mortality rate	326,120,299	326,130,673	
Women			155,066,397
Relative risk for T2D	68,535,377	235,935,399	
Incidence rate of T2D	113,541,969	202,773,705	
Prevalence rate of T2D	133,173,335	180,530,002	
Discount rate	138,639,325	174,245,651	
All-cause mortality rate	154,987,963	155,139,771	
T2D-related mortality rate	155,064,104	155,068,013	

T2D, type 2 diabetes; USD, US dollars.

## Data Availability

Data sources in this study are summarized in Table 1.

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
