# Peer review of "The Cost-Effectiveness of Increased Yogurt Intake in Type 2 Diabetes in Japan"

_nutrients, 2025, doi:10.3390/nu17142278_

Round 1
Reviewer 1 Report
Comments and Suggestions for Authors
This study discusses the positive effects of yogurt consumption on reducing T2D. The following issues need to be revised to improve the quality of the manuscript.
Line 24: "Noncommunicable diseases" are not the focus of this study, so this part can be removed.
I want to know since you repeatedly mention that multiple meta-analyses have already shown that yogurt reduces the risk of T2D. We all know that meta-analyses aggregate findings from multiple studies, which means the relationship between yogurt and T2D is already well-established. So, what is the innovative aspect of your study? I think these points could be emphasized in the introduction. Additionally, please focus the introduction on diabetes rather than noncommunicable diseases.
Line 228: "Components such as minerals, vitamins, whey proteins, and certain fatty acids found in dairy products have been suggested to reduce the risk of T2D [45]. This indicates that whey protein has insulinotropic and glucose-lowering effects." I don’t understand this sentence. Why do minerals and vitamins in dairy products reduce the risk of T2D, and how does this imply that whey protein has insulinotropic and glucose-lowering effects? What is the connection between these two sentences?
Line 232: How yogurt affects gut microbiota and subsequently regulates T2D needs to be discussed in detail.
Line 234: "Multiple meta-analyses have shown the inverse correlation between increased yogurt intake and T2D." Please cite at least 1-2 meta-analyses here.
Line 308: Why is it 4.1?
Conclusion: Please include the research significance of this paper in the conclusion.
The titles of the figures and tables are not specific enough. For example, the title of Table 2 does not highlight the main topic of the study.
Figure 1: The arrows next to elements 1 and 2 in the flowchart are too simplistic. Please use smoother lines and arrows.
Check for errors in the tables. For example, in Table 2, "Mortality" is capitalized, and "Risk" is also capitalized.
Some data formats in the tables are incorrect. For example, "1187.4–1223.8" is missing a comma. Please ensure the numbers in the tables are consistent.
The reference is a bit old
Author Response
Thank you for your constructive feedback and suggestions. We appreciate the time and effort you have put into reviewing our paper. Here’s how we plan to address your comments.
Line 24: "Noncommunicable diseases" are not the focus of this study, so this part can be removed.
Reply: We agree with your comments. We have deleted the lines as our paper specifically focuses on diabetes in Japan and it is not necessary to mention non-communicable diseases in the world. To reflect your comment, we have added the following sentences in lines 24 to 25.
Lines 24 to 25: Diabetes is a major health and economic issue in Japan, representing one of the most critical lifestyle-related diseases.
I want to know since you repeatedly mention that multiple meta-analyses have already shown that yogurt reduces the risk of T2D. We all know that meta-analyses aggregate findings from multiple studies, which means the relationship between yogurt and T2D is already well-established. So, what is the innovative aspect of your study? I think these points could be emphasized in the introduction.
Reply: Thank you for your insightful comment. As we noted in the manuscript, the preventive effect of yogurt on T2D has been well established through multiple meta-analyses. However, our study does not aim to reconfirm this association. Instead, the novelty of our research lies in evaluating the healthcare economic impact of yogurt intake as a standalone factor. To the best of our knowledge, no previous studies have specifically quantified the economic implications, such as potential reductions in healthcare costs, associated with yogurt alone. By focusing on this aspect, our study provides a new perspective that complements the existing epidemiological evidence. To clarify this point, we have added the following sentence to lines 78 to 80 of our revised manuscript.
Lines 78 to 80: Although a previous study simulated the medical economic effects of dairy intake, including yogurt, none have focused on the healthcare economic effects of yogurt alone.
Additionally, please focus the introduction on diabetes rather than noncommunicable diseases.
Reply: We agree that the introduction should focus more specifically on diabetes rather than on noncommunicable diseases in general, especially given its significance in Japan. In response, we have revised lines 29 to 34, and 36 to 41 in order to emphasize the public health importance of diabetes in Japanese context. Specifically, we added the following sentences.
Lines 28 to 33: The number of people with diabetes has shown a continuous upward trend. Diabetes is associated with complications, such as diabetic neuropathy, retinopathy, and nephropathy, and it is a risk factor for cardiovascular diseases, such as myocardial infarction and stroke. Additionally, it increases the risk of developing dementia and cancers. Furthermore, diabetes is the leading cause of new dialysis initiation.
Lines 35 to 41: Due to its profound impact on quality of life, socioeconomic vitality, and social resources, the implementation of appropriate countermeasures is important. The implementation of ongoing and comprehensive interventions to address diabetes and its associated complications is essential for promoting public health outcomes. Thus, reducing the incidence of diabetes has been designated as a key objective by the Ministry of Health, Labour and Welfare.
Line 228: "Components such as minerals, vitamins, whey proteins, and certain fatty acids found in dairy products have been suggested to reduce the risk of T2D [45]. This indicates that whey protein has insulinotropic and glucose-lowering effects." I don’t understand this sentence. Why do minerals and vitamins in dairy products reduce the risk of T2D, and how does this imply that whey protein has insulinotropic and glucose-lowering effects? What is the connection between these two sentences?
Reply: Thank you for your insightful comment. You are right to point out that the original sentence lacked clarity and logical connection between the components mentioned. While previous studies have suggested that dairy consumption was associated with a reduced risk of T2D, the specific mechanisms remain incompletely understood. Various components in dairy products such as vitamins, minerals, and whey proteins have been proposed to contribute to this potential protective effect. For example, calcium and magnesium may help improve insulin sensitivity and reduce inflammation, while vitamin D has been linked to better glucose metabolism and pancreatic beta-cell function. In particular, whey protein has been studied for its insulinotropic and glucose-lowering effects. Based on your feedback, we have revised the sentence in lines 235 to 239 to better reflect these distinctions and clarify that the components are discussed in the context of separate findings.
Lines 235 to 239: Components such as minerals, vitamins, whey proteins, and certain fatty acids found in dairy products may help to reduce the risk of T2D. Calcium and vitamin D have supportive roles in pancreatic beta-cell function and glucose metabolism. Additionally, whey protein may exert insulinotropic and glucose-lowering effects.
Line 232: How yogurt affects gut microbiota and subsequently regulates T2D needs to be discussed in detail.
Reply: Thank you for your valuable comment. We agree that the mechanism by which yogurt affects gut microbiota and subsequently influences T2D risk should be explained in more detail. To address this, we have added the following sentence in lines 241 to 246.
Lines 241 to 246: Lactic acid bacteria in yogurt reduces chronic inflammation and subsequent insulin resistance. Chronic inflammation is widely recognized as a principal factor contributing to the impairment of glucose metabolism, primarily through its role in promoting insulin resistance, which in turn facilitates the onset of metabolic disorders. The reduction of chronic inflammation through lactic acid bacteria consumption may help mitigate insulin resistance, thereby potentially improving glucose metabolism.
Line 234: "Multiple meta-analyses have shown the inverse correlation between increased yogurt intake and T2D." Please cite at least 1-2 meta-analyses here.
Reply: Thank you for your insightful comment. In response, we have added several citations after revising the sentence to lines 246 to 248.
Lines 246 to 248: Although the exact mechanisms remain unclear, multiple meta-analyses have shown the inverse correlation between increased yogurt intake and the risk of developing T2D.
Line 308: Why is it 4.1?
Reply: As per your suggestion, 4.1 was deemed unnecessary and has been removed from line 319.
Line 319: Limitations
Conclusion: Please include the research significance of this paper in the conclusion.
Reply: To reflect your comment, we have added the following sentences in the lines 348 to 352 and 356 to 357.
Lines 348 to 352: A Markov model simulation was conducted to evaluate the impact of yogurt intake on T2D and NHE in Japan. The results of this study indicate that increasing yogurt intake may serve an effective preventive strategy against T2D in Japan, with the potential to reduce both T2D incidence and NHE, underscoring the potential role of yogurt in public health.
Lines 356 to 357: This research supports future public health strategies highlighting the economic benefits of adopting dietary recommendations.
Figure 1: The arrows next to elements 1 and 2 in the flowchart are too simplistic. Please use smoother lines and arrows.
Reply: Thank you for your comment. In response, we have revised Figure 1 to improve the visual clarity of the flowchart. Specifically, we replaced the arrows with smoother and more visually refined curved arrows.
Check for errors in the tables. For example, in Table 2, "Mortality" is capitalized, and "Risk" is also capitalized.
Reply: Following your comment, we have revised Table 2.
Some data formats in the tables are incorrect. For example, "1187.4–1223.8" is missing a comma. Please ensure the numbers in the tables are consistent.
Reply: We have reviewed all the numbers and added commas where necessary in Table 2.
Reviewer 2 Report
Comments and Suggestions for Authors
The manuscript accounts of an interesting study predicting the effect of increasing yogurt consumption in Japan on the reduction of Type 2 diabetes and resulting decrease in national healthcare expenditures, using the Markov model. The results are quite convincing and may constitute a scientific argument for policy makers.
Remarks:
Numerous studies and meta-analyses have demonstrated anti-diabetic action of yogurt. The authors admit it without mentioning the possible mechanisms of this effect. In my opinion, addressing the possible mechanisms shortly could strengthen the argumentation.
Of course, the model is very approximate. The authors discuss limitations of the model and of its predictions. The sugar content of yogurt may be a factor relevant for diabetes; I expect that the input data concerned total yogurt consumption, without considering the sugar content. Non-sugar sweeteners could be considered in the recommendations, as well as the type of yogurt; low-fat, protein-rich yogurt would seem optimal.
Author Response
Thank you for your positive feedback and suggestions. We appreciate the time and effort you have put into reviewing our paper. Here’s how we plan to address your comments.
Numerous studies and meta-analyses have demonstrated anti-diabetic action of yogurt. The authors admit it without mentioning the possible mechanisms of this effect. In my opinion, addressing the possible mechanisms shortly could strengthen the argumentation.
Reply: While previous studies have suggested that dairy consumption was associated with a reduced risk of T2D, the specific mechanisms remain incompletely understood. Calcium and vitamin D have been shown to support pancreatic beta-cell function and glucose metabolism. Additionally, whey proteins in yogurt may exert insulinotropic and glucose-lowering effects. Probiotics in yogurt help maintain gut microbiota balance, which plays a key role in reducing chronic low-grade inflammation and improving insulin sensitivity. To reflect your comment, we revised sentences in lines 237 to 246.
Lines 237 to 246: Calcium and vitamin D have supportive roles in pancreatic beta-cell function and glucose metabolism. Additionally, whey protein may exert insulinotropic and glucose-lowering effects. Vitamin K2 in yogurt is inversely associated with the T2D risk. Additionally, yogurt may influence the gut microbiota through probiotics, Lactic acid bacteria in yogurt reduces chronic inflammation and subsequent insulin resistance. Chronic inflammation is widely recognized as a principal factor contributing to the impairment of glucose metabolism, primarily through its role in promoting insulin resistance, which in turn facilitates the onset of metabolic disorders. The reduction of chronic inflammation through lactic acid bacteria consumption may help mitigate insulin resistance, thereby potentially improving glucose metabolism.
Of course, the model is very approximate. The authors discuss limitations of the model and of its predictions. The sugar content of yogurt may be a factor relevant for diabetes; I expect that the input data concerned total yogurt consumption, without considering the sugar content. Non-sugar sweeteners could be considered in the recommendations, as well as the type of yogurt; low-fat, protein-rich yogurt would seem optimal.
Reply: We agree with your comment regarding the importance of yogurt composition, including sugar content and type. As you correctly noted, our simulation was based on total yogurt consumption data and did not differentiate between yogurt types or account for the presence of non-sugar sweeteners. Due to the lack of detailed consumption data, it was not possible to incorporate these variables into the model. To reflect this limitation, we have added the following sentence to lines 342 to 344 of the manuscript.
Lines 342 to 344: In addition, due to the lack of necessary data, it was not possible to incorporate variables such as yogurt type and additives like non-sugar sweeteners into the simulation.
Round 2
Reviewer 1 Report
Comments and Suggestions for Authors
The revised manuscript is fine.